# Comprehensive Insights into Spatial-Temporal Evolution Patterns, Dominant Factors of NDVI from Pixel Scale, as a Case of Shaanxi Province, China

**DOI:** 10.3390/ijerph181910053

**Published:** 2021-09-24

**Authors:** Hongliang Gu, Min Chen

**Affiliations:** School of Resources and Environment, Anqing Normal University, Anqing 246011, China; 151301001@njnu.edu.cn

**Keywords:** GIMMS NDVI3g, spatial-temporal pattern, boost regression trees, Shaanxi province

## Abstract

Based on long term NDVI (1982–2015), climate, topographic factors, and land use type data information in Shaanxi Province, multiple methods (linear regression, partial and multiple correlation analysis, redundancy analysis and boosted regression trees method) were conducted to evaluate the spatial-temporal change footprints and driving mechanisms in the pixel scale. The results demonstrated that (1) the overall annual average and seasonal NDVI in this region showed a fluctuating upward trend, especially in spring. The difference between the end of season (eos) and start of season (sos) gradually increased, indicating the occurrence of temporal “greening” across most Shaanxi Province. (2) The overall spatial distribution of annual mean NDVI in Shaanxi Province was prominent in the south and low in the north, and 98.83% of the areas had a stable and increasing trend. Pixel scale analysis reflected the spatial continuity and heterogeneity of NDVI evolution. (3) Trend and breakpoint evaluation results showed that evolutionary trends were not homogeneous. There were obvious breakpoints in the latitude direction of NDVI evolution in Shaanxi Province, especially between 32–33 °N and in the north of 37 °N. (4) Compared with precipitation, the annual average temperature was significantly correlated with the vegetation indices (annual NDVI, max NDVI, time integrated NDVI) and phenology metrics (sos, eos). (5) Considering the interaction between environmental variables, the NDVI evolution was dominated by the combined influence of climate and geographic location factors in most areas.

## 1. Introduction

The plant community composed of various vegetation is the producer in the ecosystem, which plays a stabilizing and integrating role in the overall natural environment of the land. Based on it, various ecosystems are constructed together with the animals, microorganisms and the local soil, air layer, water and other inorganic environment. [1,2]. As an important component of terrestrial ecosystems, vegetation can well reflect changes in regional ecological environment [3,4,5]. The normalized difference vegetation index is widely used in the study of vegetation cover-related activities [6,7]; this can accurately describe the vegetation growth condition and is one of the effective indicators of vegetation cover change [3,8]. The normalized difference vegetation index product of NOAA Global Inventory Monitoring and Modeling System (GIMMS NDVI_3g_, NDVI for short in this paper) is the longest time series of global vegetation data available at present; it is both reliable and accurate and is universally applicable for tracing vegetation change [9]. According to this dataset, Ye et al. [10] analyzed the change characteristics of vegetation in the global large-scale region, and the global NDVI had a distinct seasonal trend. Jiao et al. [11] studied the impact of climate change on vegetation cover in China from 1982 to 2013.

Variation in vegetation cover is strongly sensitive to climate factors. Previous studies have explored the importance of climate driving factors on vegetation cover change. Certain researchers reported that the NDVI was more sensitive to temperature than precipitation [2,11,12,13]. However, other studies revealed that the strongest relationship was between precipitation and NDVI [14,15,16,17]. In fact, natural conditions and anthropogenic factors play an important role in ecological environmental evolution [18]. Besides, few have jointly evaluated the spatial correlation between NDVI and environment drivers from the perspective of space. Moreover, previous studies have seldom quantitatively measured the interactions among various drivers. Furthermore, it is of great significance to clarify which of the natural environment and anthropogenic activities have a greater impact on plant growth for environmental protection policymaking. Redundancy analysis (RDA) can reflect the linear influence of multiple explanatory variables on response variables. For instance, Li et al. [19] used this technology to explore the main driving factors behind NDVI evolution in the Loess Plateau. Nevertheless, the relationship between environmental factors and NDVI is usually non-linear and non-stationary [20,21,22,23]. Therefore, this method has been criticized for its ability to accurately distinguish the contribution of environmental and spatial variables [24]. To overcome this criticism, the boosted regression trees (BRT) model was suggested, which can reflect the nonlinear relationship between environmental factors and satellite data [25,26,27,28,29]. Considering the spatial correlation, choosing an appropriate method to evaluate the driving factors affecting vegetation change will contribute to harnessing the potential for synergies remote sensing science and ecology.

The majority of previous studies analyzed the characteristics of temporal and spatial changes of NDVI from the perspective of large spatial scale [30,31]. However. the assessment of climate and environmental change is necessary at the sub-national scale (i.e., province) [32]. Shaanxi Province in Northwest China is one area with a fragile ecological environment which also experiences soil erosion. In turn, local human activities and social development are seriously restricted [33]. According to Shaanxi daily (http://www.shaanxi.gov.cn/xw/sxyw/202105/t20210510_2162816_wap.html, accessed on 23 August 2021), from 2000 to 2020, the increase in the percentage of vegetation index saw Shaanxi Province ranked fourth in China, and the vegetation coverage in Shaanxi Province had reached 73.29% by 2020. Although some studies have explored the relationship between vegetation cover and climate drivers in this region, the research conclusions are not consistent. For instance, Pu and Ren [34] showed that NDVI in Shaanxi province has a less significant correlation with annual temperature and precipitation. Nevertheless, Li et al. [35] demonstrated that the correlation coefficient between NDVI and climate factors (precipitation, temperature) in the same period reached a significant level. Since climate and environmental change is nonlinear and non-stationary, it is necessary to consider the impact of climate and environmental change on NDVI evolution from the perspective of a long-time series. Moreover, as an area of ecological engineering construction and reviving farmland to forest, the impact of human action on vegetation cover in this area cannot be ignored [31]. However, the interaction between natural and human factors on surface vegetation change is still unclear [36], and further research is thus required.

In addition, few studies have been conducted to quantify the effects of natural factors and human activities on vegetation indices for multivariate analysis of pixel scale in a specific area. The pixel scale analysis can accurately analyze the spatial and temporal variation of each location in the study area and the influence of driving factors in a small area, so that the spatial continuity and heterogeneity can be better reflected and analyzed. Herein, in this paper, climate factors, topography and anthropogenic activities are analyzed from multiple perspectives, and the RDA and BRT model are used to quantitatively analyze the driving force affecting NDVI from the pixel scale in order to further obtain the reasons for the dynamic changes of NDVI, provide certain data and scientific basis for the sustainable development of the ecological environment, and also provide reference for vegetation research in other regions.

## 2. Materials and Methods

### 2.1. Study Area

Shaanxi Province is located in China’s inland hinterland, with a long and narrow geographical area, and the general topography is high from north to south, low in the middle, and slopes from west to east (Figure 1a). By 2020, Shaanxi Province had a jurisdiction over 10 prefecture level cities (Figure 1b, i.e., Xi’an, Baoji, Xianyang, Tongchuan, Wei Nan, Yan’an, Yulin, Hanzhong, Ankang, Shangluo). In later sections, these names refer to the regions they contain. The geomorphic types of this area are complex and diverse, including Qinba Mountains, Guanzhong Plain and Northern Shaanxi plateau in the center of the Loess Plateau [37]. Shaanxi Province has a total area of 2.05 × 10^5^ km^2^, of which the Qinba Mountains cover 7.40 × 10^4^ km^2^, the Guanzhong Plain covers 4.94 × 10^4^ km^2^, and the Northern Shaanxi Plateau covers 8.22 × 10^4^ km^2^.

Shaanxi straddles the Yellow River and Yangtze River systems. The difference in climate between north and south is obvious. From south to north, there is a humid north subtropical climate, a warm temperate semi-humid climate and a temperate arid semi-arid climate. The annual average temperature in this province is 9–16 °C, and the annual average precipitation is 340–1240 mm. The precipitation decreases from southeast to northwest, with great differences between north and south (Figure 1c). Due to the differences in environmental factors, such as temperature, precipitation, topography and light, vegetation cover and types also vary greatly [38]. As a result of the comprehensive impact of climate characteristics, soil types and human activities, the Northern Shaanxi (such as Yulin, Yan’an) has a fragile ecological environment and experiences serious soil erosion. It is a key area for the implementation of the project of reviving farmland to forest and grassland in China. After more than ten years of development, vegetation cover has significantly changed. Xi’an, the provincial capital, is located in middle Shaanxi Province, with a long history and culture, dense population, developed economy and high urbanization. The Qinling Mountains and Daba Mountains have a great vegetation biodiversity, and are known as “biological gene bank”.

### 2.2. Datasets

The GIMMS NDVI3g dataset (http://data.tpdc.ac.cn/en/data/9775f2b4-7370-4e5e-a537-3482c9a83d88/, accessed on 12 November 2020) [39,40] were download by R gimms package [41]. The pixel size of the dataset is 1/12° × 1/12°. In order to get greater insights, the relationships in parts of the season (around max NDVI and time integrated NDVI (TI_NDVI)) [42,43] were analyzed. Meanwhile, based on the threshold methods [44], we calculated from time series annual metrics of vegetation phenology, such as start of season (sos) and end of season (eos), by using R package [45]. The monthly climate data source came from meteorological observation stations (http://data.cma.cn/, accessed on 13 January 2021) and Climatic Research Unit Time-Series version 4.03 grid data set (https://crudata.uea.ac.uk/cru/data/hrg/, accessed on 12 December 2020). The source of 90 m resolution digital elevation data was Geospatial Data Cloud (http://www.gscloud.cn/, accessed on 5 January 2021). The annual dataset of land use types with a WGS84 projection was download from https://doi.pangaea.de/10.1594/PANGAEA.913496 (accessed on 3 March 2021), which was created by Liu et al. [46]. All of the statistical analyses were performed in the R environment [47].

### 2.3. Methods

#### 2.3.1. Statistical Methods

In this paper, the linear regression model [48] was used to fit the linear change trend of NDVI pixel-by-pixel in order to obtain the spatial and temporal evolution pattern of the annual average of NDVI in Shaanxi Province. The F-test was used to test significance. Combined with the previous research [49] and F distribution table, the results were further divided into seven types (Table 1).

The study of the coefficient of variation (CV) can reflect the degree of interannual NDVI fluctuation, and a larger value of the coefficient of variation indicates a greater degree of vegetation disturbance, while a smaller value of the coefficient of variation indicates a smaller degree of vegetation disturbance and a more stable NDVI change [50].

The partial correlation analysis can analyze the correlation between NDVI and a climate factor alone without considering the influence of another factor [51]. The significance test of partial correlation coefficient was a *t*-test [48].

Multiple correlation analysis can be used to analyze the correlation between NDVI and multiple variables, and the multiple correlation coefficient of NDVI with temperature and precipitation. The significance of multiple correlation coefficient was tested by an F-test [52].

#### 2.3.2. RDA Method

RDA can quantitatively analyze the influence degree of different types of factors, which can make up for the previous studies that only analyzed the influence of various factors on NDVI [19,53]. In this paper, the R vegan package [54] was used to analyze the correlation and multiple linear regression between NDVI and environmental factors. The environmental factors were divided into three categories: climate factors (temperature and precipitation), terrain factors (altitude and slope) and geographical location factors (longitude and latitude). The larger the value is, the greater the influence of environmental factors on NDVI is. The high dependency between variables may affect the results of the RDA method. A better way to evaluate high dependency between dependent variables is to calculate the variance inflation factor (VIF). A rule of thumb is that if the VIF value exceeds 5 or 10, there is a high dependency between variables [55].

#### 2.3.3. BRT Model

The BRT model combines a boosting algorithm and a regression tree algorithm to eliminate the interaction between independent variables through multiple iterations in calculation. It has strong learning ability and is flexible in dealing with different data formats and complex data. Meanwhile, it does not need to consider the correlation between independent variables, and has a more accurate prediction ability [25,27,56]. In this paper, the model was implemented by the R dismo package [57]. The NDVI data of different land cover types [46] were used as dependent variables. The regional temperature, precipitation, altitude, slope, longitude and latitude data were used as independent variables. The relative influence value obtained by BRT model reflects the influence degree of each variable on NDVI, which can directly identify the dominant factors of NDVI in the region.

## 3. Results

### 3.1. Temporal Variation of the NDVI

#### 3.1.1. Interannual Variation

The inter-annual variation of NDVI in Shaanxi Province during 1982–2015 is shown in Figure 2. The minimum value of NDVI of whole Shaanxi Province was 0.1313, the median was 0.5033, and the mean was 0.4657. The average and median values of NDVI at all sites fluctuated and increased with time. The size of the boxplot was relatively large in some years (such as 1990, 1997), which indicated that there was a large difference between the NDVI values of each pixel in Shaanxi Province in that year. According to linear regression analysis, the average annual NDVI significantly increased at the rate of 0.0018/year (*p* < 0.001), the R-squared was 0.687, which reflected the vegetation coverage with a increasing trend in Shaanxi Province from 1982 to 2015 (Figure 3).

#### 3.1.2. Seasonal Variation

The average value of NDVI over 34 years from high to low was as follows: summer (0.5413), autumn (0.4476), spring (0.4178), winter (0.3688). The variation range of NDVI average value over all pixels in each season of every year was 0.4989–0.6150 (summer), 0.3848–0.4869 (autumn), 0.3688–0.4764 (spring), 0.2437–0.3323 (winter), respectively. The inter-annual variation trend of the average NDVI value of Shaanxi Province from 1982 to 2015 is shown in Figure 4.

The trend of NDVI fluctuated upward in all seasons in terms of inter-annual variation. The linear trend of NDVI in spring was 0.0025/year (*p* < 0.001), with the most obvious up-trend and the R-squared figure reaching 0.6919. The linear trend of NDVI in summer and autumn was cam next, reaching 0.0016/year (*p* < 0.001) and 0.0015/year (*p* < 0.001), with the R-squared reaching 0.2921 and 0.3915, respectively. The fluctuation in NDVI in winter was the least and relatively stable. The trend value of NDVI in winter was 0.0014/year (*p* < 0.001), and the increase in NDVI was the least obvious, and the R-squared value was 0.4755.

By calculating the phenological information of Shaanxi Province, we found that between 1982 and 2015, the start time of growing season was ahead, and the end time of growing season was delayed (Figure 5a). The average values of sos and eos in 1982 were 117.03 day of year (DOY) and 269.28 DOY, respectively. The corresponding values in 2015 were 82.39 DOY and 282.99 DOY, respectively. Through the overall trend of the 34-year period, the TI_NDVI value can be seen to rapidly increase before 2006 and continue to increase at a more gentle pace (Figure 5b). Max NDVI value increased slowly before 2003, but showed a rapid uptrend thereafter in Shaanxi Province.

### 3.2. Spatial Variation of the NDVI

#### 3.2.1. Spatial Distribution of Annual Mean NDVI

The spatial distribution of NDVI in Shaanxi Province between 1982 and 2015 showed a spatial distribution trend which was high in the south and low in the north, with the minimum value of 0.16 and the maximum value of 0.67 (Figure 6a).

The number of pixels with NDVI < 0.2 accounted for about 7.33% (Table 2) of the total area in Shaanxi Province, which was roughly located in the northwest of Yulin city (Figure 6a). The climate of this region belonged to temperate arid and semi-arid climate, with low precipitation and poor vegetation coverage. The number of pixels with NDVI value between 0.2 and 0.5 accounted for about 48.06% of the total area in Shaanxi Province, and was mainly distributed in the remaining areas in northern Shaanxi and central Shaanxi. Human activities were frequent in these areas, but the climate conditions were moderate. Coupled with the ecological restoration project, the vegetation coverage was relatively good. The proportion of areas with an NDVI value greater than 0.5 was 44.61%, mainly distributed in the southern part of Shaanxi Province. It belonged to the humid climate of north subtropical and warm temperate zone, and the vegetation coverage was relatively high. The image with NDVI > 0.6 was mainly located in the north and south of Hanzhong and Ankang. These areas had high altitude, and were dominated by forest land.

#### 3.2.2. Spatial Distribution of Coefficient of Variation of the NDVI

The spatial distribution of the NDVI vegetation index in Shaanxi Province not only had significant spatial heterogeneity; rather, each pixel had an obvious time change trend during 1982–2015. The CV of each pixel was calculated and is shown in Figure 6b. The overall NDVI changes during 1982–2015 in Shaanxi Province were relatively stable with a low volatility. The number of pixels with a low fluctuation (0 < CV ≤ 0.1) accounted for 82.57%, which was mainly distributed in the northwest of Yulin City, the middle and south of Yan’an City. The part with moderate fluctuation (0.1 < CV ≤ 0.15) accounted for 16.45%, which was mainly located in the east of Yulin City and the north of Yan’an City. The part of height fluctuation (CV > 0.15) accounted for only 0.98%, which was sporadically distributed in the eastern region of Yulin city.

#### 3.2.3. Spatial Distribution of Change Trend of Annual Mean and Max NDVI

The spatial distribution trends of multi-year annual average NDVI in Shaanxi Province from 1982 to 2015 were calculated, and the obtained results were reclassified (the results are shown in Figure 7a), and the number and proportion of image elements in each category of trends were further counted (Table 3).

The annual average NDVI change trend of Shaanxi Province was −0.0009–0.0034, showing an upward trend as a whole. The positive change pixels accounted for 98.83% of the total area, while the proportion of area with negative change was only 1.17%. In terms of spatial distribution, the part with a change trend of >0.003 was roughly located in the south of Ankang City, the east of Yan’an City and some areas in the north, and the upward trend of NDVI was the largest. The pixels with a change trend of 0.001–0.003 accounted for 85.71%, with a wide distribution range. The pixels between 0–0.001 accounted for 9.86%, mainly located in the central region of Shaanxi, the northwest of Hanzhong City and Yulin City, and the rise of NDVI was not obvious. The part with a change trend < 0 was mainly located in Xi’an and its surrounding areas, and NDVI showed a downward trend.

Combined with the change trend of NDVI in Shaanxi Province from 1982 to 2015 and the results of the F-test (see Figure 7b), NDVI increased significantly, accounting for the vast majority of the province, reaching 94.83%. The percentage of NDVI pixels which significantly decreased and weakly significantly decreased was 0.34%, which was consistent with the area where the trend value of NDVI was negative (i.e., the area near Xi’an City). No significant change accounted for 2.25%, which was mainly concentrated around the areas of significant decline and weak significant decline, mainly located at the junction of Xi’an, Xianyang and Baoji, and the vegetation degradation trend was not significant.

The length of the time series segments of max NDVI before the breakpoint (Figure 8a) was longer in most parts of southern and central Shaanxi, and relatively short in sporadic parts of this region and Northern Shaanxi. Trends along time terms (Figure 8c) showed that breakpoints of max NDVI occurred frequently during 2005–2010 in Shaanxi Province but were mainly concentrated in the northern area of Shaanxi. Trends along latitudes (Figure 8h) showed that there were obvious breakpoints of max NDVI value in the latitude direction over Shaanxi Province, especially between 32–33° N and in the north of 37° N.

### 3.3. Analysis of Influencing Factors of NDVI Dynamic Change

#### 3.3.1. The Temporal Relationship between the NDVI and Climate Factors

The linear trends of annual mean temperature, annual total precipitation and annual mean NDVI with year were obtained by statistical analysis (Figure 9 and Figure 10). Both temperature and NDVI showed an increasing trend, with annual mean temperature increasing significantly at a rate of 0.0449 °C/year (*p* < 0.001). The annual precipitation data fluctuated greatly and showed a downward trend at the rate of −0.2254 mm/year as a whole but failed to test as significant (*p* > 0.1). It was found that the annual average temperature significantly correlated (*p* < 0.001) with the annual NDVI, and the correlation coefficient was 0.67. Contrarily, the correlation coefficient of annual precipitation with annual mean NDVI was −0.10 (*p* < 0.5).

The correlations between seasonal precipitation and max NDVI and TI_NDVI was still not significant (Table 4). However, the relationships between temperature and max NDVI, TI_NDVI, eos and sos all reached a significant level. From the statistical analysis results (Table 4), different NDVI proxy indicators were used to correlate with the precipitation in the same period, and the results were quite different. For example, the sensitivity of TI_NDVI to annual precipitation was −0.177. However, the correlation between max NDVI and annual precipitation was 0.170.

#### 3.3.2. Spatial Relationships between the NDVI and Climate Factors

From the perspective of spatial evolution, there were great differences in the warming trend in various regions of Shaanxi Province (Figure 11). The temperature evolutionary trend in the central and western part of Yan’an City, Ankang City and the southeast corner of Hanzhong city was the smallest, and the annual warming was less than 0.3 °C. The temperature change trend in the northwest of Yulin city was the largest, and the annual temperature increase was greater than 0.5 °C. The change trend of annual total precipitation in Shaanxi Province increased from south to north, but the annual precipitation seriously decreased in Xi’an, Shangluo and Hanzhong.

The results of partial correlation between annual NDVI and temperature was −0.17–0.77, mainly positive correlation, with an average of 0.47 (Figure 12a). The percentage of pixels with positive partial correlation coefficient accounted for 99.39%, especially in Xianyang City, the middle of Baoji City, most areas of Shangluo City, Ankang City and the south of Hanzhong City. The number of negatively correlated pixels accounted for 0.61%, mainly located in the urban area of Xi’an. The results of the *t*-test on the significance of partial correlation between NDVI and temperature (Figure 12b) demonstrated that the percentage of pixels with no significance (*p* > 0.1) accounted for 15.32%, which were mainly distributed in the north central part of Yan’an City, Xi’an City, Xianyang city and Weinan City. The percentage of significant (0.01 < *p* < 0.05) and extremely significant (*p* < 0.01) pixels were 14.79% and 62.01%, respectively. There was a very significant positive correlation between NDVI and temperature in the whole province.

The mean value of partial correlation coefficient between NDVI and precipitation was 0.13, and the variation range was −0.47–0.55 (Figure 12c). The number of pixels with a positive and negative correlation coefficient was 65.23% and 34.77%, respectively. According to the *t*-test, the number of insignificant pixels of partial correlation between NDVI and precipitation accounted for 65.26% (Figure 12d), which was mainly located in the central and southern part of Shaanxi Province. The total number of pixels with significant and extremely significant results accounted for 29.82%, mainly distributed in Yulin City, the north of Yan’an City and the south of Ankang City.

#### 3.3.3. Spatial Distribution of Multiple Correlations between the NDVI and Climate Factors

NDVI is affected by the combined effect of temperature and precipitation, and the results of the multiple correlation coefficient and significance test of NDVI with pixel scale climate factors in Shaanxi Province are shown in Figure 13. The mean value of the multiple correlation coefficient of NDVI with temperature and precipitation was 0.55, and the range was between 0.19 and 0.81. The areas with low coefficient of multiple correlation were mainly located in the central part of Xi’an and Yan’an City. The high multiple correlation coefficients were concentrated in Shangluo City, Ankang City and the south-central part of Hanzhong City. The non-significant areas accounted for 7.48% and the highly significant areas accounted for 62.99%. The superimposition of the multiple correlation coefficients with the significance test results showed that the areas with low values of multiple correlation coefficients are also non-significant, which indicated that NDVI in Shaanxi Province was significantly and strongly multiple correlated with temperature and precipitation.

#### 3.3.4. Response of the NDVI to Topographic Factors

The elevation range of Shaanxi Province was 166–3738 m, and the altitude was mainly between 500 and 1500 m, accounting for 77.29% of the total area of Shaanxi Province. The mean value of NDVI first increased with the increase in altitude, and then decreased with the increase in altitude. In the range of 1000–1500 m above sea level, the mean value of NDVI reached its lowest (0.39, Figure 14). Then, with the increase in altitude, the mean value of NDVI rapidly increased and reached the highest (0.61) in the range of 2000–2500 m. When the altitude was greater than 2500 m, the mean value of NDVI tended to be stable. When the altitude was less than 500 m, the NDVI had no significant evolution trend, accounting for 1.82%. Moreover, the number of pixels with downward trend for NDVI only accounted for 0.34%, and the areas with upward trend accounted for 5.2%. In other elevation ranges, the average value of NDVI mainly increased, and the proportion without significant change was very small, showing a downward trend, and the change proportion was 0.

The slope range in Shaanxi province was between 0° and 72°, with a mean slope value of 13.35°. The slope was classified into six groups [58]. The slopes in most areas of Shaanxi Province were in the range of 0°–5°, 8°–15° and 15°–25°. A few areas were in the range of 5°–8° and >25°. With the increase in slope, the mean value of NDVI was increasing. The mean value of NDVI greatly increased in the range of 0°–25° (Figure 15). Once the slope was greater than 25°, the average growth rate of NDVI slowed downward trend. When the slope was in the range of 0°–5°, the area with no significant change in NDVI accounted for 2.05%, and the area with downward change accounted for only 0.33%. NDVI in most areas displayed an upward trend. When the slope was greater than 5°, the mean NDVI in all regions showed an upward trend.

#### 3.3.5. Spatio-Temporal Response Characteristics of the NDVI to Anthropogenic Activities

The changes in land use types reflect the influence of human activities on NDVI; thus, the statistics of land use types from 1982 to 2015 are shown in Figure 16. In 1982, the proportion of area occupied by each land use type were forest land (44.11%), barren land (26.57%), cropland land (16.35%) and grassland (12.97%). By 2015, the proportion of area in descending order was forest land (52.89%), cropland (24.23%), grassland (18.95%), and barren land (3.93%). During 1982–2015, cropland, grassland and forest land showed a fluctuating upward trend as a whole. However, the growth rate was different, and the growth rate of forest land area was the largest, increasing at the rate of 0.2771/a. The second was cropland, with a growth rate of 0.2119/a. Finally, grassland increased at the rate of 0.1438/a, and the increase was the least obvious. Only the barren land showed a decreasing trend, with an obvious downward rate of 0.6329/a. The area of land use types changed as a result of human activities, and the area change in different land use types showed that the area of forest land, grassland and cropland with high NDVI values were expanding, indicating that human activities have a positive influence on the change in NDVI.

From the perspective of spatial distribution, the barren land in Shaanxi Province was mainly concentrated in Northern Shaanxi (Figure 17). The central region of Shaanxi Province was dominated by crop land. The southern area of Shaanxi was dominated by forest land. In the time series, among the four land use types, the barren land and forest land area obviously fluctuated with time. In 1982, the area of barren land was the largest. In 1986, the area of this land type in Northern Shaanxi decreased sharply and the grassland area increased. By 1989, the barren land area increased, but decreased rapidly in 1998. Since 2002, the area of barren land had gradually decreased.

### 3.4. Relationships between Dominant Factors and NDVI under Different Land Uses

#### 3.4.1. RDA Results

The RDA method elucidated the individual and combined effects of different environmental factors on NDVI in different land use type regions, and the results are shown in Table 5. The total influence of environmental factors on NDVI values in the different land use type regions was in the order of grassland (84%) > cropland (75%) > forest land (50%) > barren land (48%). It can be seen from the table that NA values (i.e., RDA method value was less than zero) appeared to influence some environmental factors. This result may be caused by the much more complicated or nonlinear dependencies that can cause negative non-testable fractions [54,59].

In the cropland area, from a single type, the influencing factors from high to low were geographic location factors (20%), topographic factors (3%) and climate factors (2%). The combined influence of the climate and geographic location factors (31%) was the highest, much higher than the other two factors. The combined influence of three types (climate, topography and geographic location) was 19%.

In the forestland region, the individual influence of each type was in the order of topographic factors (6%) > geographic location factors (2%) > climate factors (4%). The combined influence of geographic location and climate factors was 35%. However, the combined influence of climate and terrain factor was very close to 0. The combined influence of three types was 3%.

In the grassland region, the individual influence of geographic location factors (23%) was the highest among the three types, followed by the climate factors (4%), and the topography factors (approximation 0%). The combined influence of climate and geographic location factors was 27%, and the combined influence of the three types together on NDVI was 27%.

In the barren land area, the individual influence of geographic location factors (22%) was the highest among the three types, followed by the climate factors (1%). The combined influence of topographic and geographic location factor was 14%. The combined influence of the three types on NDVI (2%) was the lowest among these four land use types.

#### 3.4.2. BRT Results

The BRT model can further identify the influence of each environmental factor on NDVI in different land use types and the corresponding changes. In the cropland area, the relative influence of predicted variables on NDVI from large to small was temperature (25.1%, Figure 18), precipitation (22.6%), latitude (19.4%), longitude (15.9%), altitude (10.2%), slope (6.9%). Partial dependence plots (Figure 18) display the average change in predicted NDVI as we varied the predicted variable while holding all other variables constant. For instance, holding all other variables constant, the predicted NDVI increased sharply with the increase in precipitation, and tended to be stable after the precipitation reached 450 mm. The predicted NDVI decreased with increasing latitude in the south of 35 °N. Between 35°N and 36 °N, it increased slightly with increasing latitude. Meanwhile, we found that the fitted function curve was very flat (i.e., close to 0), indicating that the influence of the slope on NDVI evolution was very small in the cropland area. 

In the forestland area, the relative influence of predicted variables on NDVI from large to small was temperature (22%, Figure 19), altitude (17.4%), precipitation (17.4%), slope (15.5%), longitude (14.8%), slope (12.9%). The predicted NDVI increased slowly with temperature until 13 °C. Similarity, it increased slowly with the increase in altitude. The fitted function curve of predicted NDVI was approximately a horizontal line. This result showed that the influence of slope on NDVI evolution was also very small.

In the grassland region, the relative influence of the predicted variables on NDVI from large to small was as follows: latitude (25.1%, Figure 20), precipitation (24.2%), temperature (17.7%), longitude (15.7%), slope (9.0%), altitude (8.3%). For instance, holding all other variables constant, the predicted NDVI was more stable south of 34° N, and during 34° N–36.5° N, decreased slightly and sharply and then tended to stabilize. The change curve of predicted NDVI fluctuated and rises with the increase in temperature, and stabilized after 10.5 °C.

In the barren land area, the relative influence in predicted variables on NDVI from large to small was temperature (21.3%, Figure 21), precipitation (20.3%), altitude (18.0%), slope (17.3%), latitude (12.7%), longitude (10.5%). Below 9.1 °C, the predicted NDVI decreased with the increase in temperature, then increased rapidly above 9.4 °C. The precipitation was about 355 mm, which was the dividing point for the change in the predicted NDVI. The predicted NDVI decreased sharply with the increase in altitude when the altitude was below 1200 m, and then stabilized with the increase in altitude. The predicted NDVI slowly increased with increasing slope, and continuously decreased with increasing latitude, and reached the lowest value near 39° N (Figure 21).

## 4. Discussion

### 4.1. The Homogeneity and Heterogeneity Temporal-Spatial Evolution Trend of NDVI

Temporally, Shaanxi Province showed a fluctuational increase trend in vegetation cover between 1982 and 2015, which was broadly consistent with the most parts of the global scale, especially in the mid-latitudes of the Northern Hemisphere [60,61]. After 1990, there was an obvious breakpoint in Shaanxi Province. The multi-year annual average NDVI in Shaanxi Province increased at a rate of 0.0018/year between 1982 and 2015, which was lower than the growth rate studied by Chen et al. [37], because the time period of this paper was longer and the dataset of the studied NDVI were different. From 1982 to 2015, the start time of growing season was ahead, and the end time of growing season was delayed. This result indicated temporal “greening” across most Shaanxi Province. The prolongations of the growing season also were found in high latitudes [62]. By studying the NDVI of the Inner Mongolia Plateau adjacent to Shaanxi Province, Gong et al. [63] demonstrated that the warmer spring would delay the senescence of vegetation.

Spatially, the vegetation cover in the vast majority of Shaanxi Province showed a stable increasing trend, especially in the eastern part of the Loess Plateau, which was mainly related to the project of reviving farmland to forest and grass, and the soil and water conservation project of the Loess Plateau, and the vegetation cover had significantly improved [64,65]. Trends along latitudes indicated that there were obvious breakpoints in the latitude direction of max NDVI value in Shaanxi Province, especially between 32–33° N and in the north of 37° N. Near 34° N (Xi’an City and Xianyang City), the variation trend of NDVI was the smallest. This was mainly related to the expansion of Xi’an and Xianyang City to the surrounding area to occupy part of the farmland, and the regional vegetation cover had deteriorated [1]. The length of the time series segments of max NDVI before the breakpoint was longer in most parts of southern and central Shaanxi, and relatively shorted in sporadic parts of this region and Northern Shaanxi. Trends along time terms showed that breakpoints of max NDVI frequently occurred between 2005 and 2010 in Shaanxi Province, but mainly concentrated in Northern Shaanxi. The distribution and dynamics of vegetation in Shaanxi Province had obvious spatial variation, which might have been related to the combined influence of several factors [66,67].

### 4.2. Effects of Different Factors on NDVI

Climate factors have an important influence on the growth and development of vegetation [68]. Many studies have demonstrated that temperature and precipitation were the main meteorological factors affecting the vegetation index evolution [11,61,69]. From 1982 to 2015, Shaanxi Province showed a warmer trend, which was consistent with the fact of global warming. However, the precipitation in southern Shaanxi showed a drier trend and that in northern Shaanxi revealed an upward trend. According to the average values of the whole Shaanxi Province, the annual mean NDVI was strongly positively correlated with the annual mean temperature and was slightly less sensitive to the annual mean total precipitation. From the pixel scale, partial correlation results indicated that the number of pixels with a significant correlation coefficient (*p* < 0.05) between NDVI and temperature accounted for 76.8%, and the number of pixels with significant correlation coefficient with precipitation only accounted for 29.8%.

Consequently, it can be concluded that temperature had a greater influence on NDVI evolution than precipitation in Shaanxi Province from 1982–2015. This may be because the moderate increase in temperature is conducive to improving the photosynthetic efficiency of plants and soil water use efficiency, and promoting the growth of vegetation [67]. By calculating the phenological information of this region, we also found that from 1982 to 2015, the relationships between temperature, eos and sos all reached significant level, which also showed that climate change had a great impact on vegetation growth in this region. Besides, the effect of temperature on NDVI evolution was greater after the implementation of the fallowing project [19]. Under the condition of global warming, Tucker [70] also indicated that the long-time series vegetation coverage in the middle latitudes of the northern hemisphere showed an increasing trend. From a large regional scale, most researchers demonstrated that NDVI was more sensitive to precipitation changes [16,17,71]. However, we analyzed the relationships in parts of the season (around max NDVI and TI_NDVI); the correlations between seasonal precipitation and max NDVI and TI_NDVI were still not significant. TI_NDVI and max NDVI were more sensitive to spring’s seasonal temperature.

As is known, vegetation growth is also affected by other environment factors. Topography controls the spatial redistribution of solar radiation and precipitation [72]. Shaanxi Province has complex landscape types from north to south, with large topographic undulations and different vegetation index variation characteristics at different elevations and slopes. To obtain the individual and combined influence of various environmental influences on NDVI, we conducted the RDA and BRT method. Results revealed that the combined influence of climatic factors and geographic location factors on NDVI was the highest in various land use type regions. This result was consistent with the previous research about Loess Plateau [19]. Human activities such as the implementation of reforestation and grass restoration projects had a positive effect on the restoration of vegetation [1]. Through the change in land use type area, this paper reflected the impact of anthropogenic activities on vegetation was mainly positive. A variety of influencing factors were separately analyzed in this study, but the codominant factors of the dynamic changes of vegetation cover need to be further discussed.

The BRT model can further analyze the relative influence magnitude of various influencing factors on NDVI and the variation of NDVI with the influencing factors in the region. In Figure 13, we found an interesting phenomenon that the altitude of Shaanxi Province was mainly between 1000–1500 m, but the average vegetation coverage was not the highest. Through BRT model, we indicated that altitude has little impact on forest land, crop land and grassland, but had a greater impact on barren land (Figure 21). After 1000 m, there was an obvious downward trend. From this point, it can be seen that the BRT model can explain some of the more subtle heterogeneity phenomena than the RDA method and other simple correlation statistical analysis methods can. Overall, RDA method gets the fitted values of environmental factors and NDVI after linear regression, the calculation speed is fast. The BRT model does not need to consider the interaction between environmental factors. However, this approach is highly flexible and very time-consuming for computation.

The pixel scale analysis can accurately analyze the spatial and temporal variation of each location in the study area and the influence of environment factors on a small area, so that the spatial continuity and heterogeneity can be better reflected and analyzed. From the perspective of practical application, especially the ecological restoration projects such as reviving farmland to forest and afforestation, we should study the correlation between NDVI and environmental factors on a small regional scale or even a pixel scale in order to take appropriate ecological restoration measures for different regions.

## 5. Conclusions

In this paper, we conducted on the spatial and temporal variability characteristics of NDVI (1982–2015) and its driving factors in Shaanxi Province from pixels scale using trend analysis, correlation analysis, the RDA and the BRT method. This study was meaningful for understanding the indirect response of vegetation to climate warming and anthropogenic activities at the local and regional levels.

(1)The annual mean NDVI in Shaanxi Province from 1982–2015 was 0.4361, with an overall fluctuating upward trend, increasing at a rate of 0.0018/year. The average NDVI of each season showed different degrees of increase, and the increasing trend: spring > summer > autumn > winter. The difference between start and end time of growing season increased gradually indicated that temporal “greening” across most Shaanxi Province;(2)In general, the NDVI values in Shaanxi Province demonstrated a high spatial distribution in the south and low one in the north, 98.83% of the areas indicated a stable and increased trend of annual average NDVI in Shaanxi Province in past 30 years, and only 1.17% of the areas demonstrated a decreasing trend of multi-year annual average NDVI. Pixel scale analysis showed that there was spatial continuity and heterogeneity in NDVI changes in the study area;(3)In terms of temporal variation, the correlation coefficients between NDVI metrics (annual mean NDVI, max NDVI, TI_NDVI, eos, sos), and temperature all reached significant level. However, the evolutionary trends of NDVI in this study area were not sensitive to precipitation. The results of spatial correlation analysis showed that 76.80% of the study areas showed a significant correlation with temperature, and only 29.82% significantly correlated with precipitation. The NDVI values were partially decreasing at elevations below 500 m and slopes in the range of 0°–5°, while the rest mostly increased;(4)The results of RDA and BRT method showed that the combined influence of climatic and geographic location factors was the greatest in most land use type regions, and temperature may be the dominant factor in NDVI evolution dynamics in grass land area.

In this paper, the influence of human activities on NDVI was only considered from the change in land use type area based on the GLASS-GLC-7 data set [46], which had only four land use types (crop land, forest, grass land, barren land) in Shaanxi Province. As is known, vegetation growth is affected not only by temperature and precipitation, but also by population density, light duration, soil humidity and other factors. Therefore, in future research, according to the ecological data of ground-based observations, we will also include these factors in more comprehensive research.

## Figures and Tables

**Figure 1 ijerph-18-10053-f001:**
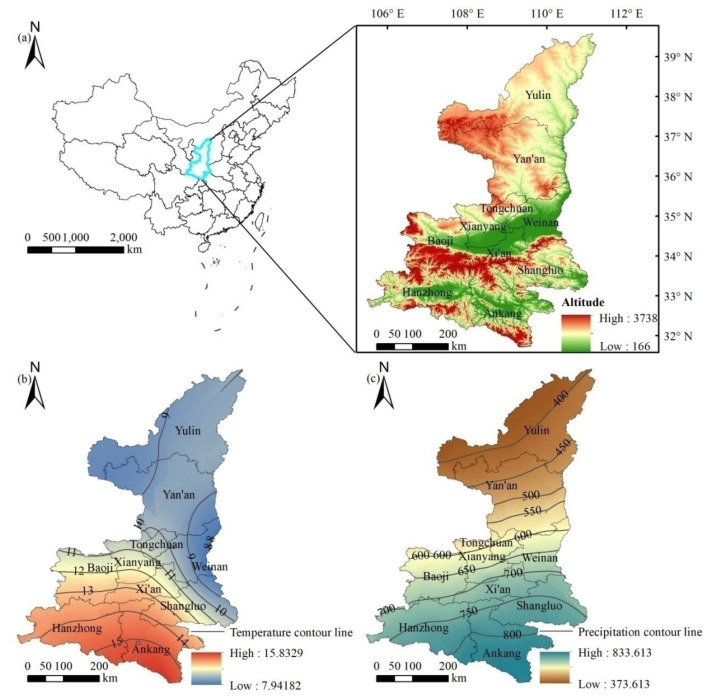
Geographical location, geographical boundaries, and the overview of physical geography (dem, temperature, and precipitation) of the study area (The dotted lines in left panel of (**a**) displays the geographical boundaries of China, the right panel of (**a**) displays altitude of Shaanxi province; (**b**) is the spatial distribution of annual average temperature (the units is °C); (**c**) is the spatial distribution of annual average precipitation (the units is mm).

**Figure 2 ijerph-18-10053-f002:**
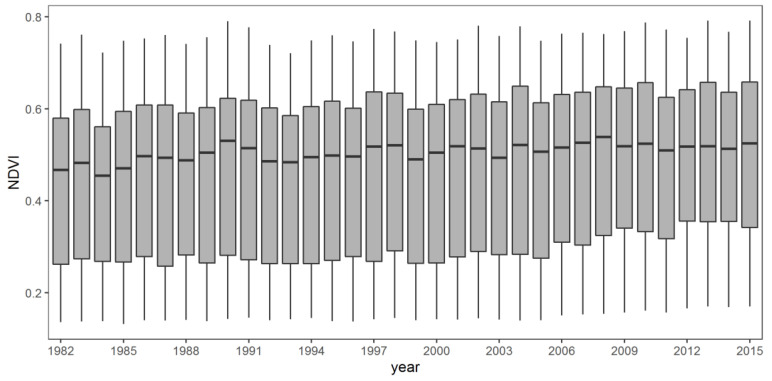
The boxplot of inter-annual variation of NDVI in Shaanxi Province during 1982–2015 (Each box denotes data distribution represents the NDVI data distribution of the whole Shaanxi Province in that year. The line that divides the box into 2 parts represents the median of the data. The end of the box shows the upper (75%) and lower (25%) quartiles).

**Figure 3 ijerph-18-10053-f003:**
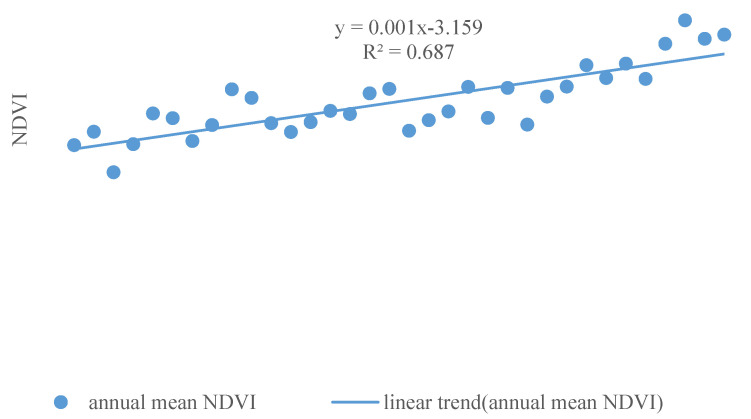
Inter-annual variation of annual mean NDVI in Shaanxi Province during 1982–2015 (The *x*-axis represents the year).

**Figure 4 ijerph-18-10053-f004:**
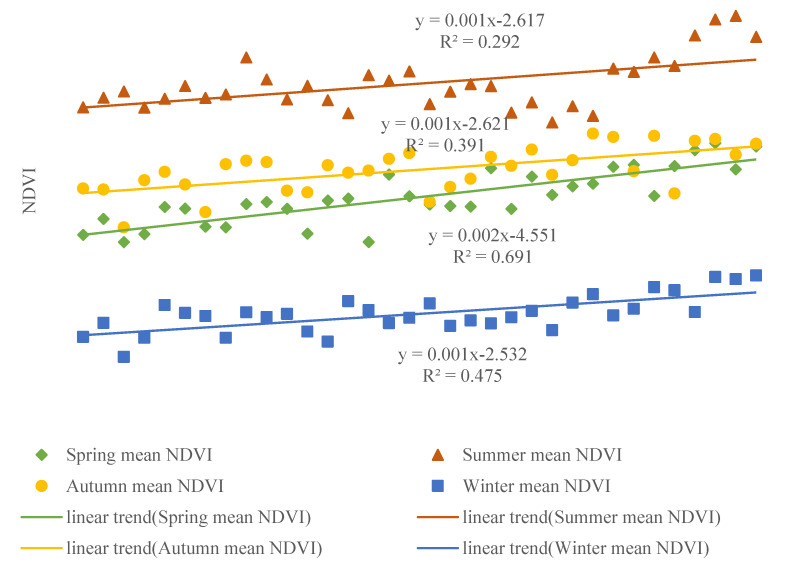
Inter-annual variations of seasonal mean NDVI in Shaanxi Province during 1982–2015 (Spring (March–May), Summer (June–August), Autumn (September–November), Winter (December, January, February). The *x*-axis represents the year).

**Figure 5 ijerph-18-10053-f005:**
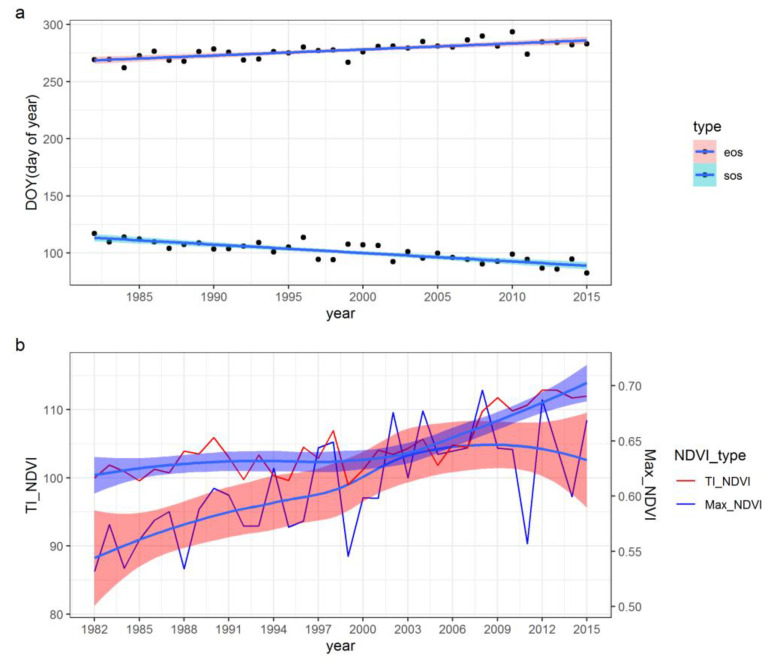
The annual metrics of vegetation phenology (**a**) and the season (max NDVI and time integrated NDVI) time series (**b**) (“sos” denotes start of season, “eos” denotes end of season, “Max_NDVI” represents max NDVI, “TI_NDVI” represents time integrated NDVI).

**Figure 6 ijerph-18-10053-f006:**
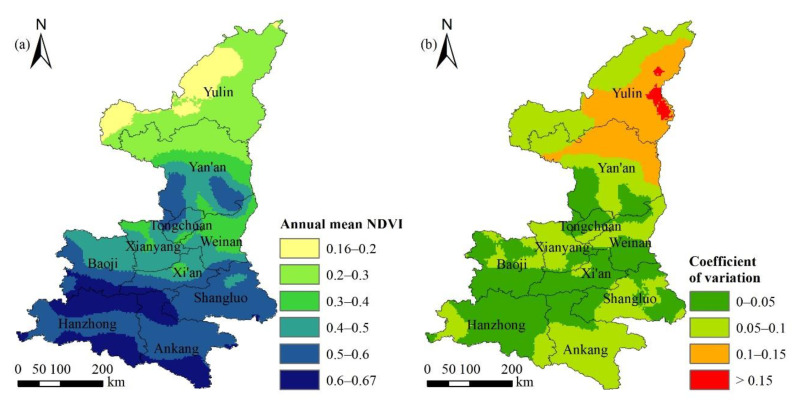
Spatial distribution characteristics of multi-year mean NDVI (**a**) and coefficient of variation (**b**) in Shaanxi Province during 1982–2015.

**Figure 7 ijerph-18-10053-f007:**
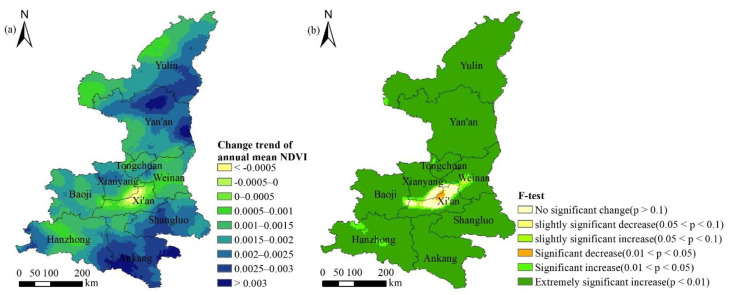
Spatial distribution of change trend of annual mean NDVI (**a**) and its significance (**b**) in Shaanxi Province during 1982–2015.

**Figure 8 ijerph-18-10053-f008:**
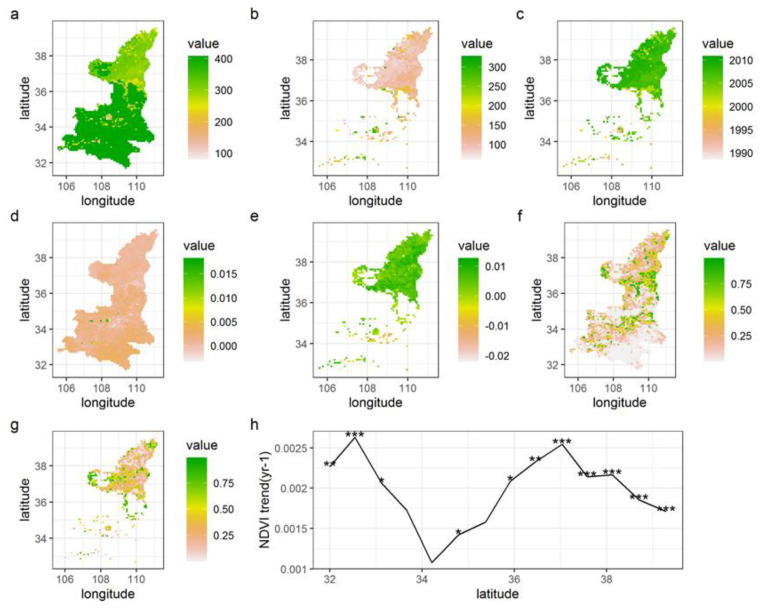
The trends and breakpoints on max NDVI time series from pixel scale (**a**): the length of the time series segments before the breakpoint (LSEG1); (**b**): the length of length of the time series segments after the breakpoint (SEG1); (**c**): breakpoints; (**d**): the slope of SEG1; (**e**): the slope of SEG2; (**f**): the *p*_value of the SEG1; (**g**): the *p*_value of SEG2; (**h**): latitudinal gradient of NDVI trends. *** *p* < 0.001; ** *p* < 0.01; * *p* < 0.05. The ‘latitude’ label on the axis represents north latitude, and the ‘longitude’ label on the axis represents the east longitude).

**Figure 9 ijerph-18-10053-f009:**
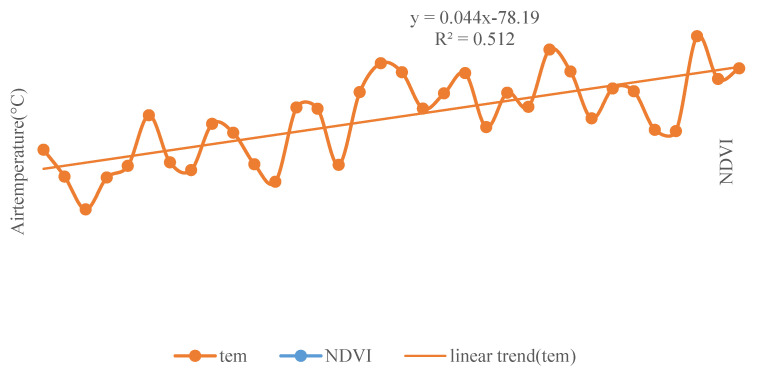
Variation curve of NDVI and annual mean temperature in Shaanxi Province during 1982–2015 (“tem” denotes the temperature. The *x*-axis represents the year).

**Figure 10 ijerph-18-10053-f010:**
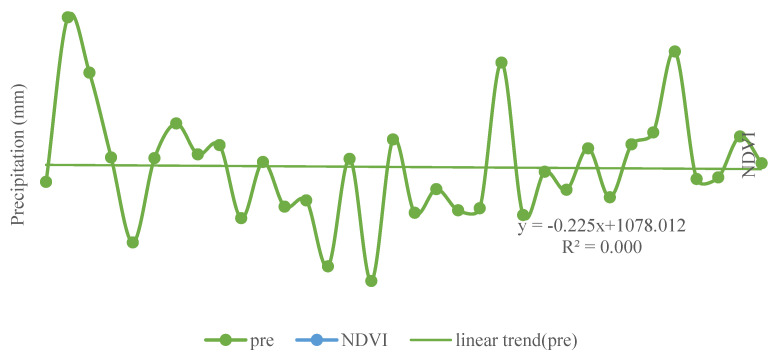
Variation curve of NDVI and annual mean precipitationin Shaanxi Province during 1982–2015 (“pre” denotes the precipitation. The *x*-axis represents the year).

**Figure 11 ijerph-18-10053-f011:**
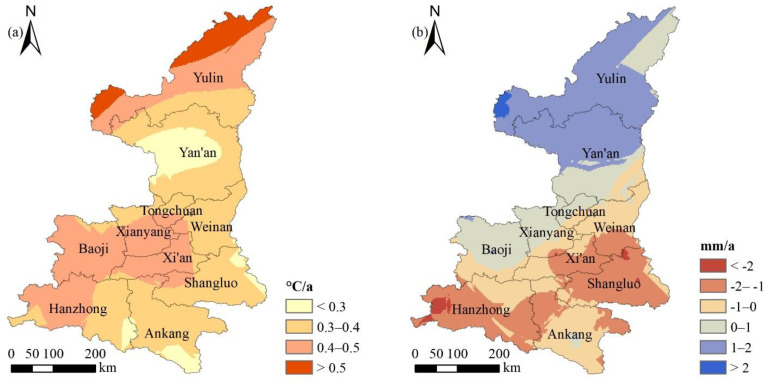
Spatial distribution of temperature (**a**) and precipitation (**b**) evolution trends over time in Shaanxi Province during 1982–2015.

**Figure 12 ijerph-18-10053-f012:**
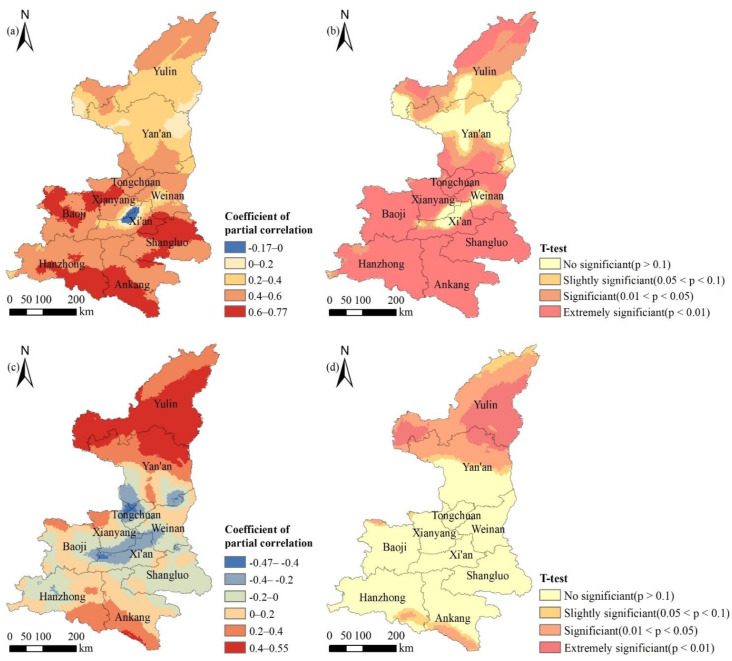
Spatial distribution of the partial correlation coefficients and its *t*-test results between NDVI and temperature (**a**): coefficient of partial correlation; (**b**): *t*-test result, precipitation; (**c**): coefficient of partial correlation; (**d**): *t*-test result in Shaanxi Province.

**Figure 13 ijerph-18-10053-f013:**
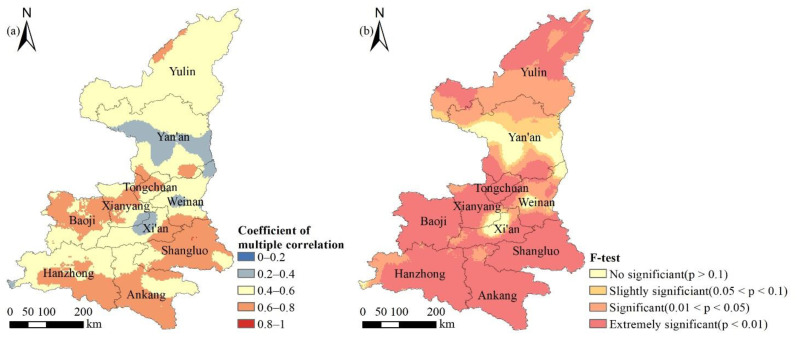
Spatial distribution of the multiple correlation coefficients (**a**) and its F-test results (**b**) between NDVI and temperature, precipitation in Shaanxi Province.

**Figure 14 ijerph-18-10053-f014:**
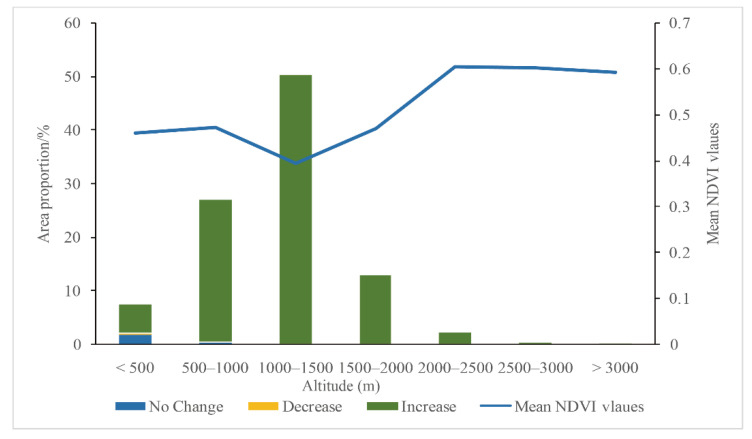
The relationship of NDVI change with elevation in Shaanxi Province (Note: “No Change”, ”Decrease”, and “Increase” in the legend means the annual average NDVI change trend of Shaanxi Province, referring to the Figure 7).

**Figure 15 ijerph-18-10053-f015:**
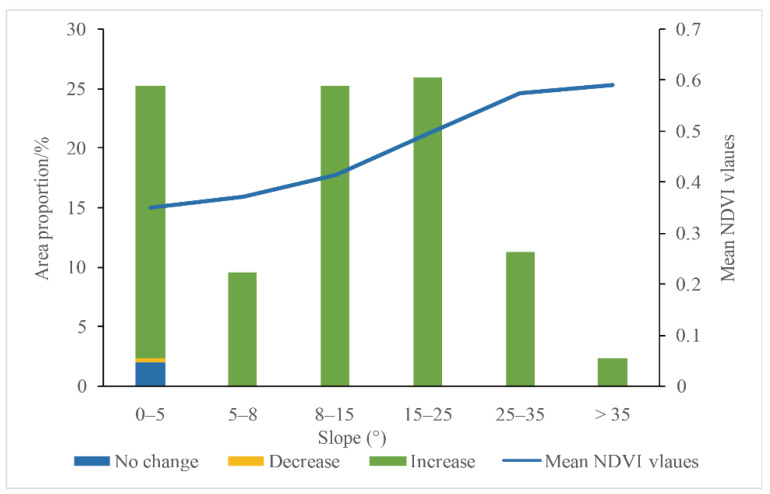
The relationship of NDVI change with slope in Shaanxi Province (Note: “No Change”, ”Decrease”, and “Increase” in the legend means the annual average NDVI change trend of Shaanxi Province, referring to the Figure 7).

**Figure 16 ijerph-18-10053-f016:**
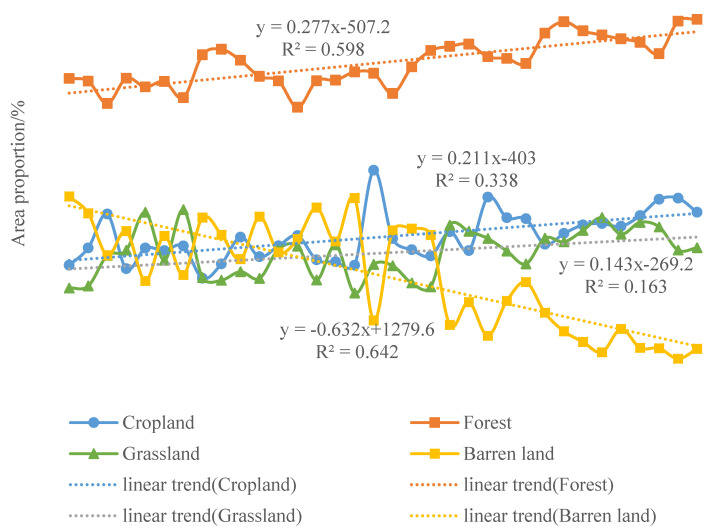
The trends of area proportion changes for each landcover type during 1982–2015.

**Figure 17 ijerph-18-10053-f017:**
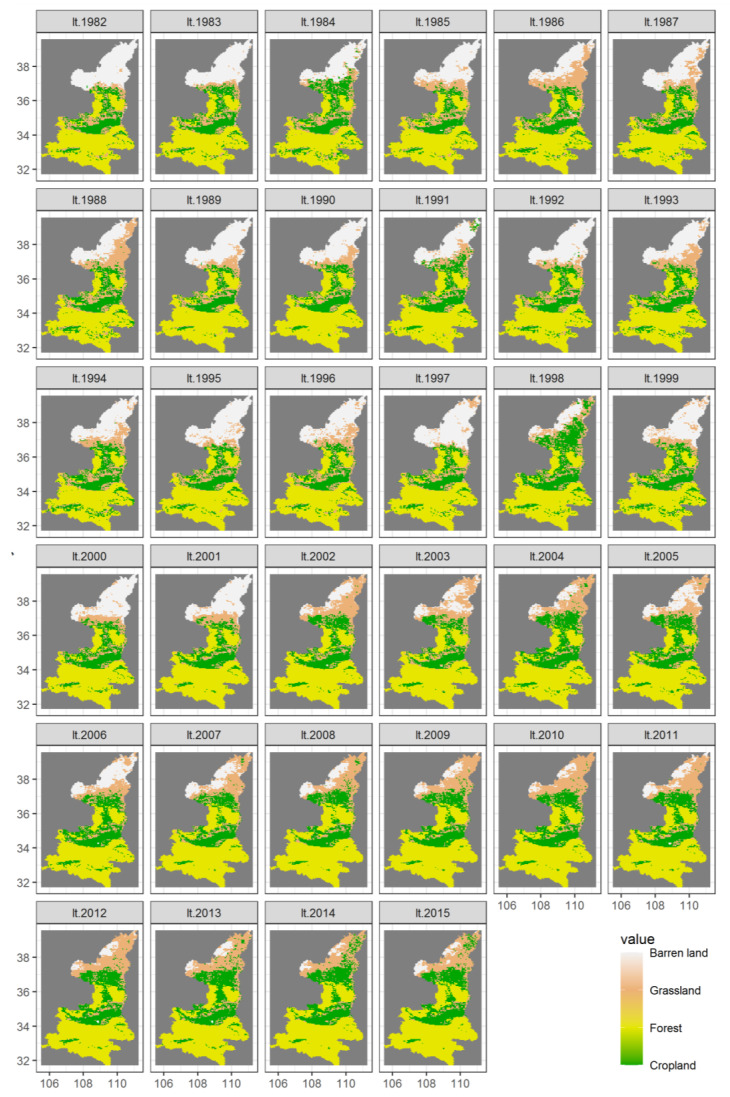
A spatial set of maps showing land cover during 1982–2015 (“lt.1982” denotes the land cover in 1982, and so on).

**Figure 18 ijerph-18-10053-f018:**
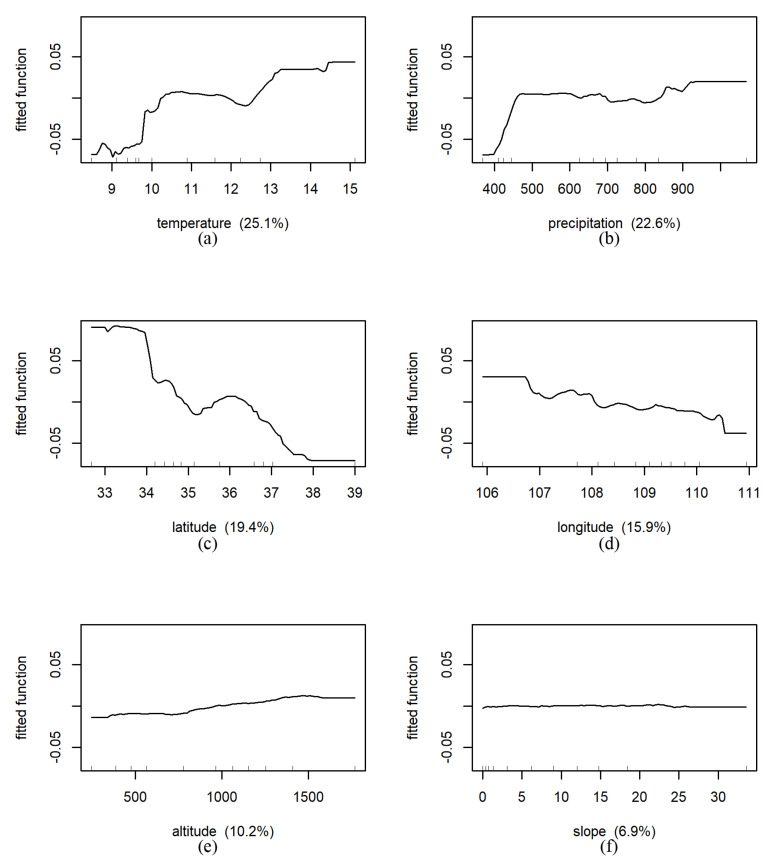
Results of marginal effects of environmental factors on NDVI under cropland in whole Shaanxi Province (The fitted function of NDVI vs. (**a**): temperature (°C), (**b**): precipitation (mm), (**c**): latitude (N), (**d**): longitude (E), (**e**): altitude (m), (**f**): slope (°), respectively. The *x*-axis denotes the predictive variable; the contents in brackets in the title of the *x*-axis represent relative influence. The *y*-axis denotes the marginal effect of the selected variables by “integrating” out the other variables).

**Figure 19 ijerph-18-10053-f019:**
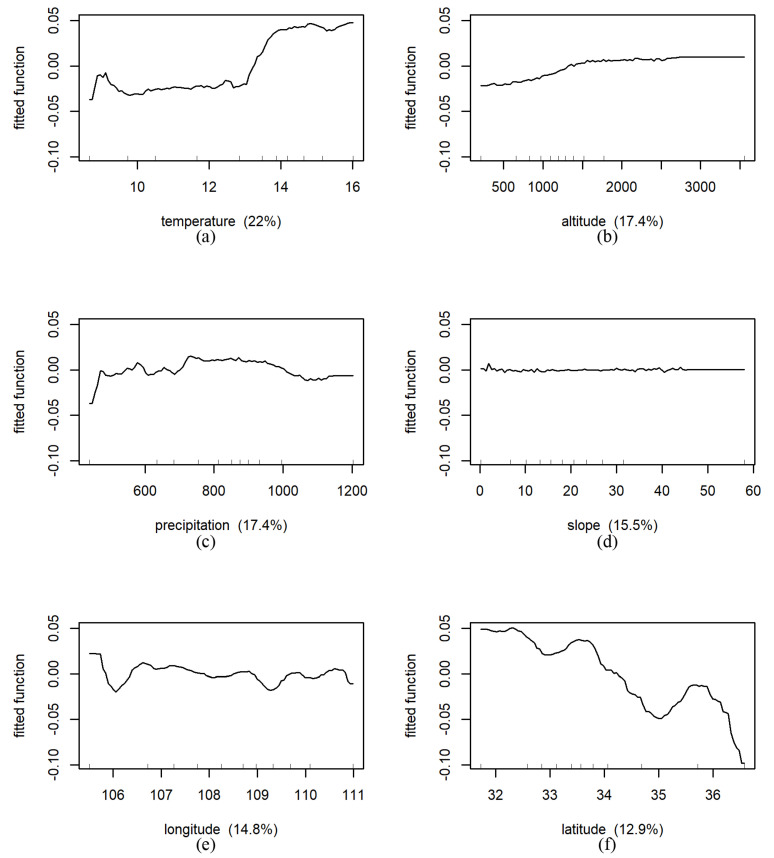
Results of marginal effects of environmental factors on NDVI under forest land type in whole Shaanxi Province (The fitted function of NDVI vs. (**a**): temperature (°C), (**b**): altitude (m), (**c**): precipitation (mm), (**d**): slope (°), (**e**): longitude (E), (**f**): latitude (N), respectively. The *x*-axis denotes the predictive variable; the contents in brackets in the title of the *x*-axis represent relative influence. The *y*-axis denotes the marginal effect of the selected variables by “integrating” out the other variables).

**Figure 20 ijerph-18-10053-f020:**
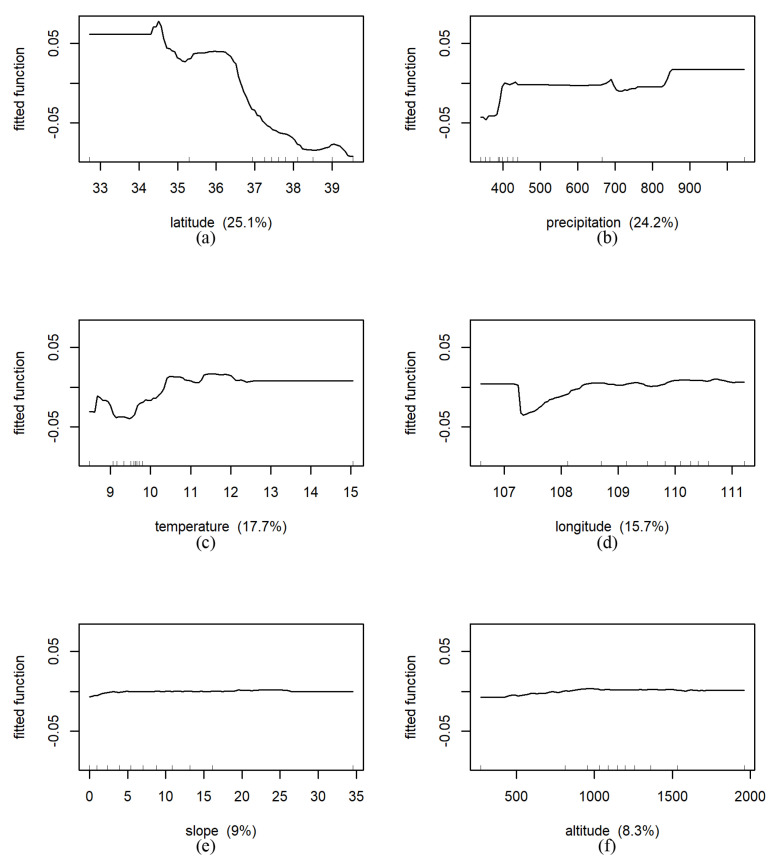
Results of marginal effects of environmental factors on NDVI under grassland in whole Shaanxi Province (The fitted function of NDVI vs. (**a**): latitude (N), (**b**): precipitation (mm), (**c**): temperature (°C), (**d**): longitude (E), (**e**): slope (°), (**f**): altitude (m), respectively. The *x*-axis denotes the predictive variable; the contents in brackets in the title of the *x*-axis represent relative influence. The *y*-axis denotes the marginal effect of the selected variables by “integrating” out the other variables).

**Figure 21 ijerph-18-10053-f021:**
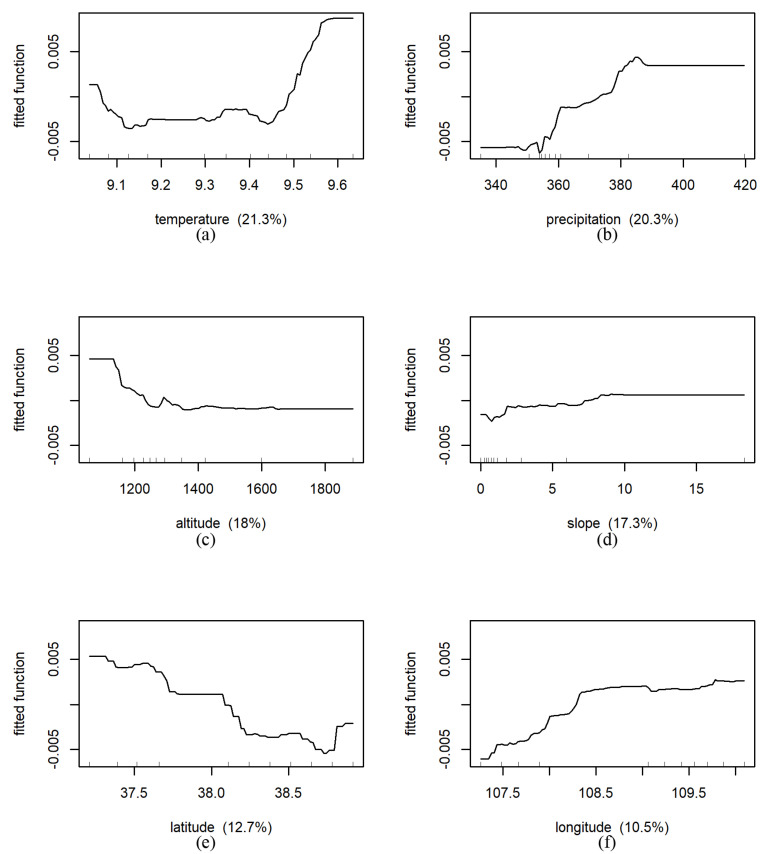
Results of marginal effects of environmental factors on NDVI under barren land in whole Shaanxi Province (The fitted function of NDVI vs. (**a**): temperature (°C), (**b**): precipitation (mm), (**c**): altitude (m), (**d**): slope (°), (**e**): latitude (N), (**f**): longitude (E), respectively. The *x*-axis denotes the predictive variable; the contents in brackets in the title of the *x*-axis represent relative influence. The *y*-axis denotes the marginal effect of the selected variables by “integrating” out the other variables).

**Table 1 ijerph-18-10053-t001:** Different types of NDVI change trend and significance test (F-value: the significance of F-test; *p*-value: the significance of linear regression model; b: the slope of linear regression model. When b > 0, the annual NDVI value shows an upward trend; when b < 0, the annual NDVI value shows a downward trend).

Typesof Change	F-Value	*p*-Value	b
No significant change	F < 2.869	*p* > 0.1	
Slightly significant decrease	2.869 ≤ F < 4.149	0.05 < *p* < 0.1	b < 0
Slightly significant increase			b > 0
Significant decrease	4.149 ≤ F < 7.499	0.01 < *p* < 0.05	b < 0
Significant increase			b > 0
Extremely significant decrease	F ≥ 7.499	*p* < 0.01	b < 0
Extremely significant increase			b > 0

**Table 2 ijerph-18-10053-t002:** Number of pixels and proportion of different types of multi-year mean NDVI in Shaanxi province during 1982–2015.

Multi-Year Mean NDVI Values	Number of Pixels	Proportion/%
0.16–0.2	1948	7.33
0.2–0.3	5610	21.10
0.3–0.4	2369	8.91
0.4–0.5	4800	18.05
0.5–0.6	8154	30.67
0.6–0.67	3706	13.94

**Table 3 ijerph-18-10053-t003:** Pixel numbers and proportion of change trend of the NDVI in Shaanxi province during 1982–2015.

Change Trend	Pixel Numbers	Proportion/%
<−0.0005	135	0.51
−0.0005–0	176	0.66
0–0.0005	298	1.12
0.0005–0.001	2325	8.74
0.001–0.0015	6593	24.80
0.0015–0.002	6841	25.73
0.002–0.0025	5537	20.83
0.0025–0.003	3800	14.29
>0.003	883	3.32

**Table 4 ijerph-18-10053-t004:** The relationships between NDVI metrics and climate factors (sos: start of season, sos: end of season. *** *p* < 0.001; ** *p* < 0.05; * *p* < 0.1).

Climate Factor	TI_NDVI	Max NDVI	eos	sos
annual temperature	0.651 ***	0.419 **	0.600 ***	−0.714 ***
spring seasonal temperature	0.687 ***	0.380 **	0.524 ***	−0.685 ***
summer seasonal temperature	0.505 **	0.358 **	0.532 ***	−0.571 ***
autumn seasonal temperature	0.301 *	0.381 **	0.360 **	−0.421 **
winter seasonal temperature	0.374 **	0.140	0.342 **	−0.400 **
annual precipitation	−0.177	0.170	−0.117	0.102
spring seasonal precipitation	0.06	0.196	−0.108	−0.152
summer seasonal precipitation	−0.156	−0.022	−0.051	0.238
autumn seasonal precipitation	−0.191	0.148	−0.096	0.042
winter seasonal precipitation	0.175	0.083	0.184	−0.078

**Table 5 ijerph-18-10053-t005:** Influence of environmental factors on NDVI on the four land cover types using RDA method (X1 represents climatic factors (temperature, precipitation), X2 denotes topographic factors (altitude, slope, aspect), X3 represents geographic location factors (longitude, latitude), X1 ∩ X2 denotes the interaction influence between climatic factors and topographic factors, and so on. “NA” denotes the values < 0, “0.0” denotes the values are very close to 0).

	Cropland	Forest	Grassland	Barren
X1	3.0	4.0	4.0	1.0
X2	3.0	6.0	0.0	7.0
X3	20.0	2.0	23.0	22.0
X1 ∩ X2	NA	0.0	0.0	NA
X1 ∩ X3	31.0	35.0	27	2.0
X2 ∩ X3	NA	NA	3.0	14.0
X1 ∩ X2 ∩ X3	19.0	3.0	27	2.0

## Data Availability

Not applicable.

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
