# Peer review of "Comprehensive Insights into Spatial-Temporal Evolution Patterns, Dominant Factors of NDVI from Pixel Scale, as a Case of Shaanxi Province, China"

_ijerph, 2021, doi:10.3390/ijerph181910053_

Round 1

Reviewer 1 Report

Summary Statement

This paper explores the covariability of NDVI from 1982-2015 with climate variables, topographic factors and land use type in Shaanxi province in China. They find that NDVI is increasing primarily in spring and that temperature is more important than precipitation in determining NDVI while altitude and slope also have a bearing on NDVI.

One of the more informative types of research is to understand the factors that influence NDVI variability and change in a given location. It allows us to understand the forcing mechanisms and this paper follows along these lines of inquiry. I think this is a publication worthy paper but after major revision.

I think the paper could be shortened to sharpen its message. There were many extra things in here that did not really add to the main points (e.g., Hurst analysis). The paper is fairly clearly written but would benefit from additional editorial input and some structuring of the paragraphs (in discussion in particular). I have provided some suggestions in the minor edits. There is also a good deal of methodology in the results section that makes the reading difficult. As I read the paper, I felt that too many technical methods were applied and I think in revising the paper, the authors should chose only those that are most important to telling their story. Some methods seem to provide duplicative results and take up too much room in the manuscript. Are both RDA and Boosted Tree needed? My other major comment is that the analysis was done using annual data. The NDVI is very low in winter and since it is a measure of the growing season, I think the results would be more clear if the focus is on the growing season. So, I suggest that the authors reduce what is presented and dig a bit deeper into the interpretation for those things that are included.

Major

1) Introduction: Currently the introduction is missing a brief paragraph that describes the vegetation of this region. It is always nice to see a map of land cover to orient oneself.

2) Materials and methods: Two panels with temperature and precipitation climatology may be nice to add Figure 1. It could also be later with a climatological plot of NDVI in the region.

3) The Hurst Exponent (Lines 136-144 + Section 3.2.4), The use of the Hurst exponent here is not adding anything to the paper and is not really correct. First, this is a very short time series (34 years) and Hurst analysis typically needs 100 years of climate data for a meaningful result as it needs a minimum of 2 decades of data. I believe the data were not detrended before the Hurst exponent was calculated so it will naturally suggest persistence because there is a trend in this data. This material does not add any value to the paper and is a distraction.

4) Captions. Overall most of the captions should provide more information. The captions should state that the correlations/analysis is at the annual scale (or seasonal). In general, the captions should better describe the figures.

5) Analysis period, Is there any real value in looking at the annual mean relationships? I think the paper should focus on the growing season. The annual relationships could be mentioned but I think it complicates the interpretation. The correlations between annual precipitation is not significant (is this going to be true if you just look at the growing season?) and this may hide a seasonal relationship. I still think analyzing the relationships in parts of the season (around max NDVI and time integrated NDVI) may give you greater insight. In the Arctic the time integrated NDVI (defined as the sum of the biweekly NDVI above a threshold of 0.05 over the growing season) is best correlated with climate over Maximum NDVI.

6) Section 3.3.4, The analysis shown in Figure 13 with topography is very interesting. I think it could benefit in interpretation to include vegetation types and precipitation and temperature to explain why this is the case. I think this is worth expanding on to provide some reasons why this is the case in Shaanxi province.

7) Figure 14, Does the direction of the slope matter? South facing or north facing?

8) Figure 15, It would be nice to see a spatial set of maps showing land cover for different years along with this time series.

9) Section 3.4.1, I think it would be easier for the reader to look at the numbers in a table which would make it easier to compare different landcover and their prevalent environmental factors. This section is very hard to absorb the way it is currently presented.

10) Section 4, The discussion should be tightened up. It has a lot of material but the story is not clear to me the way the way it is currently written.

11) The authors are missing a key citation for the GIMMS3g paper: Pinzon J and Tucker C 2014 A non-stationary 1981–2012 AVHRR NDVI3g time series Remote Sens. 6 6929–60

Minor

1) abstract line 21, the use of ‘greater’ suggest there is something else which this influence is greater than but it is not listed. It is better to replace greater with ‘dominates’.

2) line 36, change to ‘longest time series of global vegetation data’

3) Figure 1 left panel, what is the dashed line in the bottom right of the panel? It is not explained in the caption. The caption also needs to discuss each panel. For example: “The left panel displays the geographical boundaries of China (may be good to include other countries for reference in this panel). The right panel displays altitude of Shaanxi province.” I do not believe geomorphology is the correct term for what is shown, since the processes that went into making this landscape are not given in the graphic.

4) line 182 remove ‘.’

5) line 206, why is the minimum value -0.274? There are flags in the NDVI3g which are negative but by definition NDVI should be between 0 and 1. As the paper says this is bare land or more likely water (should find that pixel ) which maybe is not of interest for this paper and can be removed once it is clarified that it is water. Could it be missing data?

6)  Section 3.2.2, All readers may not be familiar with the geography of China so I was confused by the use of Yulin City. Does that refer to the regions shown in Figure 5, right panel? A definition that this refers to a region would be helpful because when I see the word city, I looked for a dot on the map to show where the city is located. When the first map is introduced, the authors can tell the reader how the different regions will be discussed in the paper to orient them to the geographical divisions. I am thinking they are called ‘City’ from reading further in the paper, so all you have to do is define it early on so the reader is ready.

7) Figure 6. I assume this is NDVI change per year given these magnitude values. That should be clearly noted in the caption.

8) Line 295, what does it mean that the vegetation has improved? Is there ground truth to support this? So far the analysis shows that the NDVI, a measure of photosynthetic activity of plants is going up. The implication is that the plants have more biomass but it is a given without some ground truth.

9) line 296, what does yuan in the middle of this sentence mean?

10) line 332, correct sentence structure.

11) line 349-353, What is a bias correlation coefficient?

12) line 424, remove ‘.’

13) line 600-601, ‘Shaanxi Province has a large north-south span’ for statements like this it is always good to quantify by giving the latitudional range in parentheses.

14) Figure 4 does not define the months that go into the seasons.

Author Response

Dear reviewer, thank you for reviewing my manuscript in your busy schedule and giving me the opportunity to modify and promote it.

Reviewer 2 Report

The article deals with with the establishment of the relationship between NDVI and environmental factors. The topic is worthy of research. In the current state of the manuscript, it is well written, the methodology is detailed and the results are clear. However, the discussion section needs to be enriched as it does not discuss the implications of the results found and their comparison with previous studies. Some minor comments/changes need to be addressed in order to be considered for publication.

General comments

I have also provide numerous suggestions for improving the analysis.

The main concern is that the novelty of the research is not fully clear, since such issue has been already developed in several previous manuscript (also in not quoted literature). If such novelty is not clearly highlighted, the risk is that the manuscript looks more a simple case study rather than a research paper.

What are the new contributions that differ from previous studies?

What are the advantages and disadvantages of the methodology used in this study?

What are the limitations of this study? They should be carefully mentioned in the discussion section.

How does the study help bridge science and practice? Authors could improve the interest to the readers by highlighting what knowledge gap they are filling with their paper

*The answer to these questions should be reflected in the manuscript*

Specific comments

Line 31-32: Put the abbreviation NDVI and use in the rest of the introduction, e.g. Line 34-35.

Line 35-36: Font size is different, edit.

Line 73: Change "Fig.1" to "Figure 1". Throughout the manuscript in the same way.

Line 80: Put a "." before "There".

Line 91: Remove the X and Y axis of the left and bottom grid, it is repetitive. Remove the blue frame.

Line 91: What do the dotted lines indicate?

Line 94: Please state manufacturer, city and country from where software has been sourced. Provide the download link of the NDVI3g dataset.

Line 94: What is the pixel size?

Line 96: Provide the web link.

Line 99: Provide the link to the cloud data.

Line 99: Referring to ArcGIS, please state manufacturer, city and country from where software has been. What version is it?

Line 125: Remove a "."

Line 138: Provide the equation.

Line 185: Please state manufacturer, city and country from where software has been sourced.

Line 187-189: Did you test if there is any high dependency between these variables (e.g. correlate temperature with altitude?). This may affect the results of the RDA method.

Line 189: A "." is missing.

Line 197: Please state manufacturer, city and country from where software has been sourced.

Line 197: What are these various types of land uses?

Line 241: What method of Kirging was employed: Ordinary, CoKriging, etc.?

Line 535: Where is the citation?

Line 613: Does this indicate evidence of climate change? How can it be related?

Author Response

(The authors gave the same response as above.)

Round 2

Reviewer 1 Report

Insights into spatial-temporal patterns, dominant factors from the complex relationship between NDVI and environmental proxies

Hongliang Gu and Min Chen

Summary

The manuscript has improved through sharpening its focus and presentation. I think the key findings are also more clearly conveyed. There are a few things that still need attention.

First, the manuscript needs some editorial attention. I provided more science related edits in the minor comments but additional grammar and punctuation edits are needed. Part of this editorial change should work on structuring the text in each paragraph in a clear progression. Because currently the paper is not easy to read and follow. I found much of it confusing and I have read the paper before. I have tried to give some specific comments below in Major and minor comments. One additional comment that I have is that the paper is quite long now and I expect in the editorial changes the text can be tightened.

Major

1) Introduction: The content and structure looks good in the intro,  but there are numerous typographic and language errors, so I suggest you ask an English language expert to provide edits. I have listed some below in the minor comments but more work is needed to smooth out the text.

2) Methods: I am not sure equations 1-6 are necessary and currently for 4-6 not all the variables are defined. These are standard methods and a reference could direct the reader to the equation.

3) General comments about the study.

- Figure 2, interesting that it looks like over time the spread decreases of the NDVI over the study region. The boxes appear to be getting smaller to me.

- The criteria used to define start and end of season should be clearly described. The text says this is done in the R package but what is the criteria? Because this is a tricky quantity to pinpoint using the AVHRR NDVI.

- Please remove technical information from the results section, for example lines 364-368, line 468-469, line 642-644 (check for more instances). If you need the technical information it should not be the first sentence someone reads in a paragraph.

- please review the concept of  ‘topic sentence’ to make that the first sentence of each paragraph. The topic sentence tells the reader what the point is of the paragraph and the rest of the text is in support of this topic sentence. This paragraph in section 3.2.1 should start with a sentence that describes the trend of NDVI. ‘The spatial mean annual NDVI displays a pattern of … …’ Then details can be provided. A topic sentence structure will help the reader understand your message.

-

Minor

1) line 14, what is eos and sos?

2) line 16, change ‘improving’ to ‘increasing’

3) line 19, meaning unclear  of ‘reverse continuous change’

4) line 10, RDA and BRT should be defined.

5) line 71, remove ‘.’ at beginning.

6) line 72, change ‘has obvious seasonal’ to ‘has a clear seasonal’

7) line 113, change ‘coer’ to ‘cover’

8) line 156, ‘exuberant vegetation’ should be replace by ‘great vegetation biodiversity’ , which is what I think this sentence means to convey, but I am not positive.

9) Figure 1 caption need units for T and P, ËšC and mm. please check other captions for units (elevation units in. figure 14).

10) line 203, ‘It’ should be removed.

11) line 241, ‘Similarly’ should not be capitalized.

12)line 322, use of ‘improvement’ is not appropriate because a higher NDVI is what is meant, unless I am missing something regarding revegetation described as an improvement. Improvement implies a value judgement when the findings should describe what is happening to the NDVI, going up or going down?

13) figure 2 caption, replace ‘inner’ with ‘inter’. Also the caption should signify the top and bottom of the boxes plus the line (75% and 25% and median, I presume).

14) lines 375-376, what does ‘wind and sand area’ mean? Sand dunes?

15) Table 2 caption ‘pixel numbers’ should be ‘Number of pixels’. It is somewhat subtle but if you say pixel numbers then it suggests something different than the count of pixels, which is what I think you mean to convey. Maybe just replace pixel numbers with ‘pixel count’ to avoid this confusion.

16) line 675, missing caption for this figure?

Author Response

Dear reviewer, thank you for reviewing my manuscript in your busy schedule. At the same time, thank you for giving me the opportunity to modify and improve the quality of the manuscript. According to your suggestion, I have carefully revised it, learned a lot of knowledge, and improved my scientific research writing ability. According to your comments, I asked a native English speaker to help me check the grammar and spelling. If necessary, I will use English Editing Services to provide language services at a later stage.
